# Prognosis of amyotrophic lateral sclerosis with cognitive and behavioural changes based on a sixty-month longitudinal follow-up

**Shan Ye**[1,2], **Pingping Jin**[1,2], **Lu Chen**[1,2], **Nan Zhang**[1,2], **Dongsheng Fan**[1,2]*

**1** Neurology Department, Peking University Third Hospital, Beijing, China, **2** Beijing Municipal Key Laboratory of Biomarker and Translational Research in Neurodegenerative Diseases, Beijing, China

* dsfan2010@aliyun.com

**Data Availability Statement:** All relevant data are within the manuscript and its Supporting Information files.

## Abstract

### Objective

Approximately 50% of amyotrophic lateral sclerosis (ALS) patients have cognitive and behavioural dysfunction in varying degrees and forms. Previous studies have shown that cognitive and behavioural changes may indicate a poor prognosis, and cognitive function gradually deteriorates over the course of disease, but the results of different studies have been inconsistent. In addition, there are relatively limited long-term follow-up studies tracking death as an endpoint. The purpose of this study was to investigate the clinical prognostic characteristics of ALS patients with cognitive behavioural changes through long-term follow-up in a cohort.

### Methods

A total of 87 ALS patients from 2014 to 2015 in the Third Hospital of Peking University were selected and divided into a pure ALS group, an ALS with behavioural variant of frontotemporal dementia (ALS-bvFTD) group, and an ALS with cognitive and behaviour changes group. All patients were followed up for 60 months. The main end point was death and tracheotomy.

### Results

There was no significant difference in survival curve between pure ALS and ALS with cognitive and behavioural change group, but the survival time of ALS-bvFTD group was significantly lower than the other two groups (P < 0.001). For those who was followed up to the endpoint, the survival time of the ALS-bvFTD group was significantly shorter than that of the pure ALS group (t = 5.33, p < 0.001) or the ALS with cognitive and behaviour changes group (t = 4.25, p < 0.001). The progression rate of ALS Functional Rating Scale–Revised (FRS-R) scores from recruitment to endpoint was significantly faster in the ALS-bvFTD group than in the pure ALS group (z = 2.68, p = 0.01) or the ALS with cognitive and behavioural changes group (z = 2.75, p = 0.01). There was no significant difference in survival time (t = 0.52, P = 0.60) or FRS-R score progression rate (z = 0.31, p = 0.76) between the pure ALS group and the ALS with cognitive and behavioural changes group. The total Edinburgh Cognitive and

**Funding:** This work was partially supported by grants from National Natural Science Foundation of China (Project No. 82001350), Peking University Third Hospital Key Clinical Projects (Project No. BYSY2018048), Peking University Third Hospital Cohort Construction Project (Project No. BYSYDL2019002), and The National Key Research and Development Program of China (Project No.2018YFC1315201).

**Competing interests:** The authors have declared that no competing interests exist.

Behavioural Amyotrophic Lateral Sclerosis Screen (ECAS) score was positively correlated with survival time (r = 0.38, p = 0.01).

## Conclusion

ALS-bvFTD patients have shorter survival time. The total ECAS score may be correlated with survival time.

## Background

Amyotrophic lateral sclerosis (ALS) is a neurodegenerative disease with high disability and mortality rates. The clinical heterogeneity of this disease is very strong, and patients with different phenotypes have very different prognosis. In recent years, increasing attention has been paid to cognitive and behavioural changes in ALS. Up to 50% of ALS patients may have different degrees and forms of cognitive and behavioural dysfunction. Five percent of patients even fulfil the diagnostic criteria for frontotemporal lobe degeneration (FTLD), with the behavioural variant of frontotemporal dementia (bvFTD) being the most common type [1–5]. Therefore, in 2017, Strong et al. proposed the concept of amyotrophic lateral sclerosis frontotemporal spectrum disease (ALS-FTSD) [2]. Previous studies have shown that patients with cognitive and behavioural changes may have a poor prognosis [6–8]. Cognitive function deteriorates gradually over the course of the disease [9–12]. However, the results of different studies have been inconsistent [13, 14]. In addition, there are relatively limited long-term follow-up studies tracking death and tracheotomy as endpoint. The purpose of this study was to investigate the clinical prognostic characteristics of ALS patients with cognitive and behavioural changes through 60-month follow-up in a cohort.

## Materials and methods

### 1. Participants

A total of 87 ALS patients who met the revised El Escorial diagnostic criteria [15] from 2014 to 2015 in Peking University Third Hospital were selected. The exclusion criteria included the following: (1) other central nervous system diseases, such as Parkinson's syndrome, cerebrovascular disease, brain trauma, or epilepsy; (2) illiteracy; (3) severe physical dysfunction, rendering the subject completely unable to cooperate with neuropsychological examination. The clinical information and neuropsychological Edinburgh Cognitive and Behavioural Amyotrophic Lateral Sclerosis Screen (ECAS) scores of 84 of these ALS patients have been published previously [16]. As reported in this article, one patient met ALS-bvFTD criteria according to Rascovsky [17] and Strong criteria [2]. Forty-two patients with normal ECAS score and no behavioural symptom were defined as pure ALS group. The other forty-one patients with abnormal ECAS score and/or behavioural symptoms but can't meet Strong's ALS-FTD criteria [2] were defined as ALS with cognitive and behavioural change group. There were three ALS-bvFTD patients enrolled during the same period. The three patients together with the one reported, were defined as ALS-bvFTD group. The clinical information of 87 ALS patients was shown in Table 1. All patients were followed up for 60 months, during which telephone follow-up was given every half year, until loss to follow-up or getting the endpoint. The main endpoint was death and tracheotomy.

During the 60 months, 42 of 87 ALS patients reached the endpoint, among which 3 patients were taken tracheotomy, 27 patients died of respiratory failure, 1 patient died of multiple

**Table 1. Clinical information of 87 ALS patients.**

| | Pure ALS (n = 42) | ALS with cognitive and behavioural changes group (n = 41) | ALS-bvFTD (n = 4) |
|---|---|---|---|
| Gender | | | |
| Male | 30 | 27 | 4 |
| Female | 12 | 14 | 0 |
| Age (M±SD) | 52.90±11.16 | 56.98±9.95 | 63.50±10.34 |
| Education (years) (M±SD) | 12.62±3.50 | 10.15±2.89 | 12.50±4.04 |
| Duration of illness (months) (M±SD) | 16.45±14.76 | 15.32±14.60 | 15.75±8.42 |
| Onset of disease | | | |
| Bulbar | 8 | 8 | 3 |
| Cervical | 24 | 24 | 1 |
| Thoracic | 0 | 3 | 0 |
| Lumbar | 10 | 6 | 0 |
| Diagnostic level | | | |
| Definite | 7 | 5 | 1 |
| Probable | 14 | 11 | 1 |
| Laboratory-supported probable | 12 | 13 | 1 |
| Possible | 9 | 12 | 1 |
| FRS-R at enrolment (M±SD) | 41.55±5.49 | 41.02±3.66 | 40.25±4.42 |
| Total ECAS score (M±SD) | 98.10±10.90 | 75.41±15.04 | 56.75±40.41 |

FRS-R: Revised Functional Rating Scale; ECAS: Edinburgh Cognitive and Behavioural Amyotrophic Lateral Sclerosis Screen.

organ failure, and the cause of 11 patients were unknown; 22 ALS patients received early follow-up, but refused further follow-up or could not be contacted before the endpoint; 23 ALS patients (26.44%) were loss to follow-up.(Fig 1)

This study was approved by the Research Ethics Committee of Peking University Third Hospital. All participants signed the consent before included by themselves or guardians as set forth by the Declaration of Helsinki. Those who had physical dysfunction that couldn't write or severe cognitive and behavioural dysfunctions, consents were signed by their guardians. The consent procedure was approved by ethics committees.

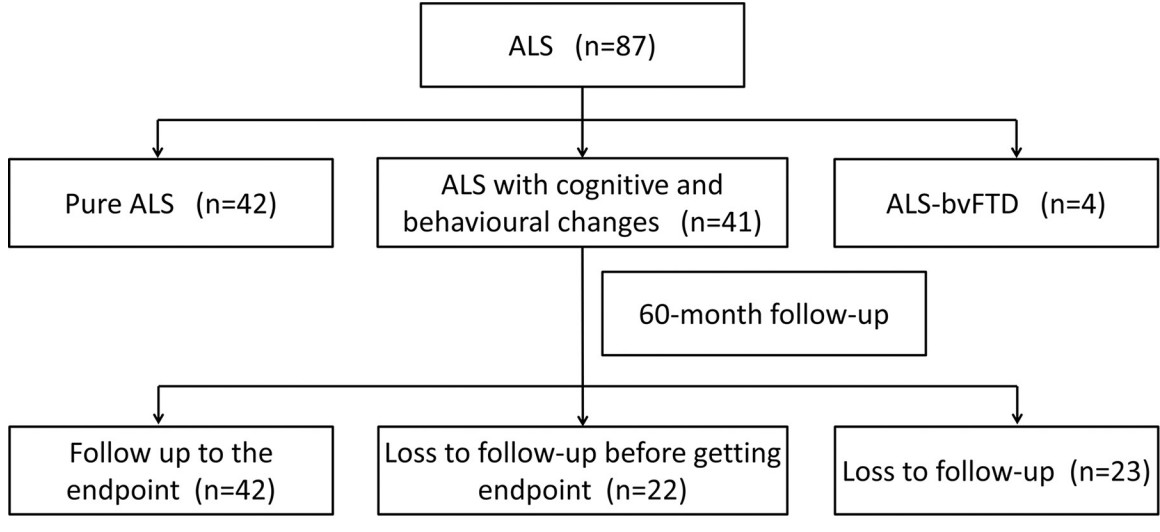

**Fig 1. Flow of participants through the study.**

## 2. Statistics

There were three subgroups in this study. When we compared age, years of education, duration of illness, total ECAS score, Functional Rating Scale–Revised (FRS-R) score at enrolment, FRS-R progression rate from onset to enrolment, FRS-R progression rate from enrolment to endpoint, and survival time between each pair of subgroups, a t-test was used for those who met a normal distribution, and the nonparametric Mann-Whitney U test was used for those who did not. The chi-square test was used to compare the distributions of gender, disease onset and diagnostic level between the two groups. When we analysed the correlation between the ECAS score and FRS-R score, FRS-R progression rate from onset to enrolment, FRS-R progression rate from enrolment to endpoint, and survival time, the Pearson method was used when the two groups of variables were linearly correlated; otherwise, the Spearman method was used. Kaplan-Meier survival curve was used for survival analysis. A threshold of $p<0.05$ was used to define significant differences. Statistical analysis was performed using SPSS 18.0 statistical software.

## Results

### 1. Survival analysis of ALS patients with and without cognitive and behavioural changes

When taking the Kaplan-Meier survival analysis, the survival curve of pure ALS group and ALS with cognitive and behavioural change group had no difference, while the survival time of ALS-bvFTD group was significant shorter ($p<0.001$). (Fig 2) Log Rank test was used to compare the survival curve of pure ALS group and ALS with cognitive and behavioural change group, there was no significance ($p = 0.79$).

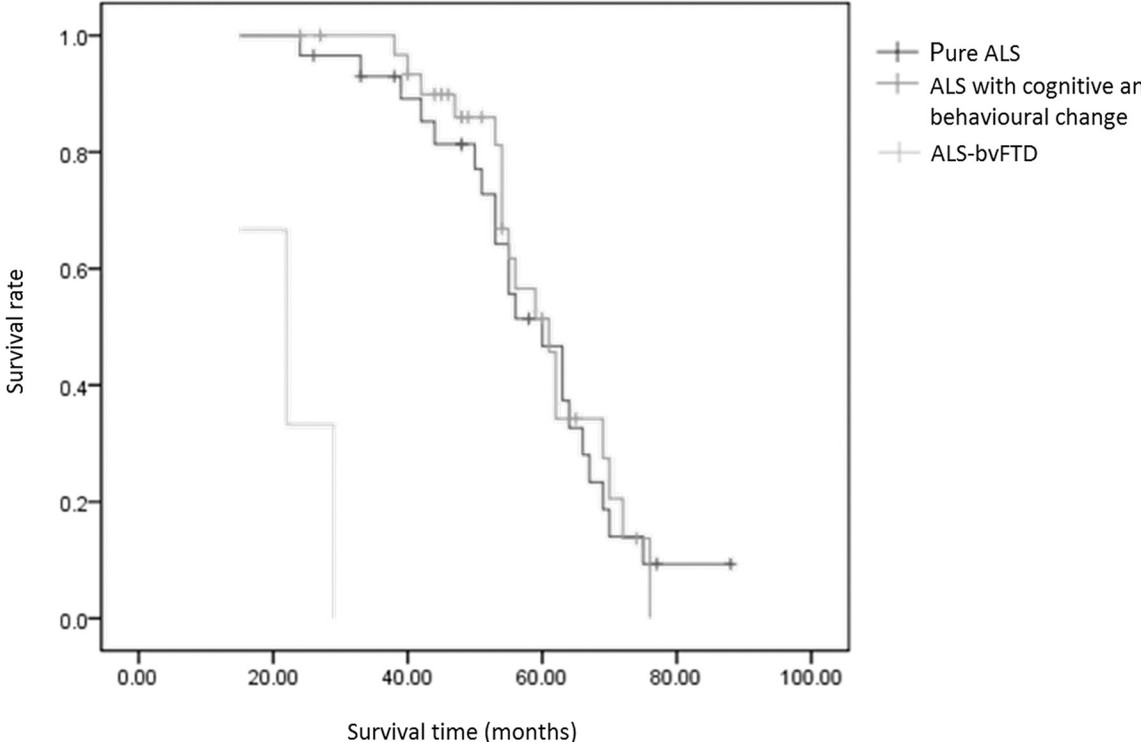

**Fig 2. Survival analysis curve of ALS patients.**

**Table 2. Comparison between followed up to the endpoint and not to the endpoint groups.**

| | Followed up to the endpoint (n = 42) | Not followed up to the endpoint (n = 45) | p |
|---|---|---|---|
| Group | | | 0.41 |
| Pure ALS | 18 | 20 | |
| ALS with cognitive and behavioural | 31 | 24 | |
| ALS-bvFTD | 3 | 1 | |
| Gender | | | 0.82 |
| Male | 30 | 31 | |
| Female | 12 | 14 | |
| Age (M±SD) | 56.55±10.11 | 54.16±11.38 | 0.30 |
| Education (years) (M±SD) | 11.00±3.15 | 11.87±3.67 | 0.29 |
| Duration of illness (months) (M±SD) | 16.10±16.13 | 15.69±12.64 | 0.39 |
| Onset of disease | | | 0.52 |
| Bulbar | 9 | 10 | |
| Cervical | 21 | 28 | |
| Thoracic | 2 | 1 | |
| Lumbar | 10 | 6 | |
| Diagnostic level | | | 0.78 |
| Definite | 10 | 5 | |
| Probable | 12 | 14 | |
| Laboratory-supported probable | 13 | 14 | |
| Possible | 7 | 12 | |
| Total ECAS score (M±SD) | 80.60±18.87 | 90.09±19.30 | **0.02**[*] |
| FRS-R (M±SD) | 40.24±4.34 | 42.18±4.73 | 0.05 |

[*]p<0.05.

## 2. Comparison between followed up to the endpoint and not to the endpoint groups

Except for ECAS score, there was no significant difference between the ALS patients who followed up to the endpoint group and not to the endpoint group (Table 2). The percentage of followed up to the endpoint patients was much higher in ALS-bvFTD (3/4), and ECAS score of ALS-bvFTD patients are usually lower. So when we got rid of the ALS-bvFTD patients, there was no difference of ECAS score between the two groups (t = 1.55, p = 0.12).

## 3. Analysis of ALS patients followed up to the endpoint

**3.1 Clinical information of ALS patients followed up to the endpoint.** The clinical information of ALS patients who was followed up to the endpoint is shown in Table 3.

Among these patients, the pure ALS group had more years of education than the ALS with cognitive and behavioural changes group (z = 2.60, p = 0.01), and the onset site of the ALS-bvFTD group was significantly different from those of the pure ALS group ($\lambda^2 = 8.75$, p = 0.01) and the ALS with cognitive and behavioural changes group ($\lambda^2 = 10.29$, p = 0.02); other than those differences, however, there was no difference in gender, age, years of education, duration of illness, disease onset or diagnostic level between any two subgroups (p > 0.05).

The total ECAS score in the pure ALS group was significantly higher than that in the ALS with cognitive and behavioural changes group (t = 7.07, p < 0.001) or ALS-bvFTD group

**Table 3. Clinical information of the ALS patients followed up to the endpoint.**

| | Pure ALS (n = 18) | ALS with cognitive and behavioural changes group (n = 21) | ALS-bvFTD (n = 3) |
|---|---|---|---|
| Gender | | | |
| Male | 14 | 13 | 3 |
| Female | 4 | 8 | 0 |
| Age (M±SD) | 55.05±10.66 | 57.05±9.57 | 62.00±12.12 |
| Education (years) (M±SD) | 12.33±3.12 | 9.81±2.71 | 11.33±4.04 |
| Duration of illness (months) (M±SD) | 19.22±20.88 | 13.14±11.70 | 18.00±8.72 |
| Onset of disease | | | |
| Bulbar | 3 | 3 | 3 |
| Cervical | 9 | 12 | 0 |
| Thoracic | 0 | 2 | 0 |
| Lumbar | 6 | 4 | 0 |
| Diagnostic level | | | |
| Definite | 2 | 7 | 1 |
| Probable | 4 | 7 | 1 |
| Laboratory-supported probable | 8 | 4 | 1 |
| Possible | 4 | 3 | 0 |
| Total ECAS score (M±SD) | 95.44±8.27 | 73.81±10.49 | 39.00±23.64 |
| FRS-R at enrolment (M±SD) | 40.50±5.41 | 40.24±3.45 | 38.67±3.79 |
| FRS-R progression rate from onset to enrolment (M±SD) | 0.58±0.44 | 0.80±0.52 | 0.57±0.30 |
| Percentage of patients followed up to the endpoint | 42.86% | 51.22% | 75.00% |
| FRS-R progression rate from enrolment to endpoint (M±SD) | 1.05±0.41 | 1.12±0.48 | 17.90±16.05 |
| Survival time (months) (M±SD) | 56.89±10.82 | 54.86±12.97 | 22.00±7.00 |

(t = 8.26, p < 0.001). The total ECAS score in the ALS with cognitive and behavioural changes group was significantly higher than that in the ALS-bvFTD group (t = 4.59, p < 0.001).

**3.2 The characteristics of progression prognosis in different subgroups.** There was no significant difference in FRS-R score or FRS-R progression rate from onset to enrolment between the pure ALS group, the ALS with cognitive and behavioural changes group and ALS-bvFTD group (p > 0.05).

The survival time of the ALS-bvFTD group was significantly shorter than that of the pure ALS group (z = 2.72, p = 0.002) or the ALS with cognitive and behavioural changes group (z = 2.66, p = 0.008) (Fig 3).

There was no significant difference in the FRS-R progression rate from enrolment to endpoint between the pure ALS group or the ALS with cognitive and behavioural changes group (z = 0.31, p = 0.76). The FRS-R progression rate from enrolment to endpoint in the ALS-bvFTD group was significantly higher than that in the pure ALS group (z = 2.68, p = 0.01) or the ALS with cognitive and behavioural changes group (z = 2.75, p = 0.01) (Fig 4).

**3.3 Correlation between cognitive score and disease progression index.** There was a positive correlation between the total ECAS score and survival time (r = 0.38, p = 0.01), but there was no significant difference in the total ECAS score and FRS-R score at enrolment, the FRS-R progression rate from onset to enrolment, or the progression rate from enrolment to endpoint (p > 0.05) (Fig 5). The results were similar when using cubic and other regression models. As for the ECAS subscores, language (r = 0.35, p = 0.02) and ALS-specific function score (r = 0.42, p = 0.01) had positive correlation with survival time, while the others didn't (p > 0.05).

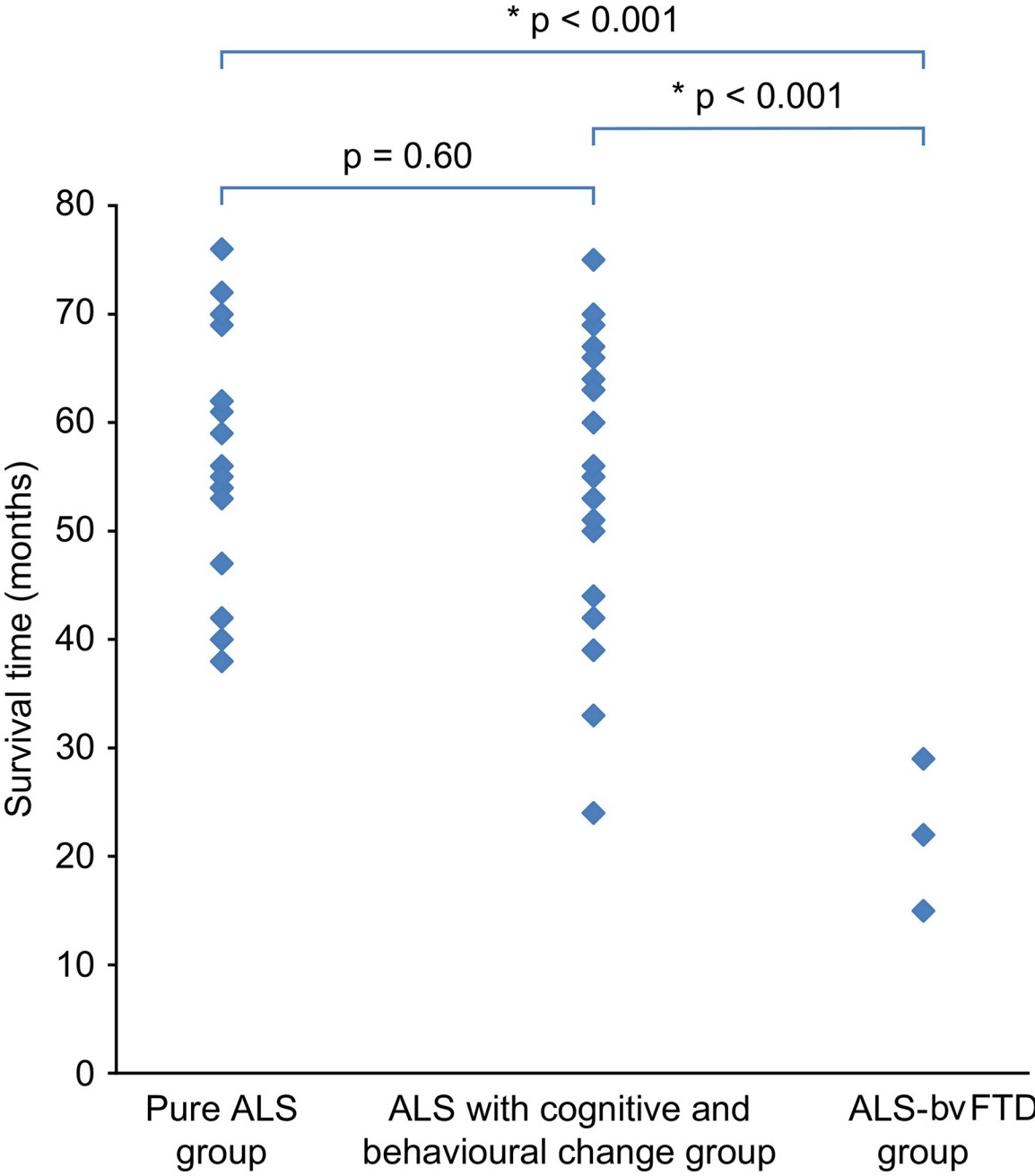

**Fig 3. Comparison of survival time among subgroups.**

## Discussion

The results showed that the survival time of ALS-bvFTD patients was significantly shortened, which was consistent with the previous literature. Floeter et al. found that the survival rate of ALS-bvFTD patients with the C9orf72 gene mutation after 12 months and 18 months of follow-up were significantly decreased [12]. Govaarts et al. also found that the survival time of patients with frontotemporal lobe symptoms was significantly shortened [18]. Bersano's study even showed that ALS patients who had worsening of their cognitive function had worse

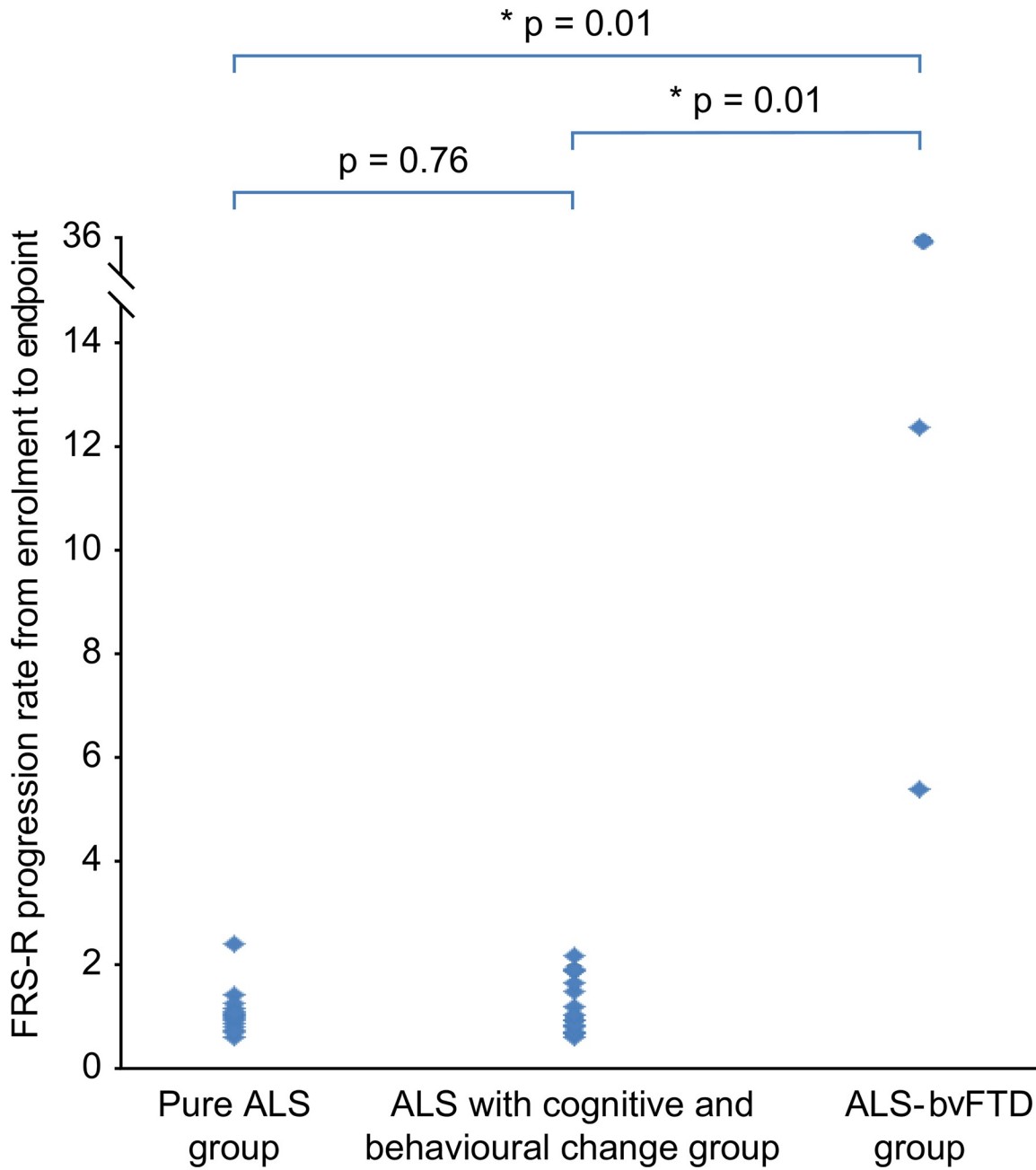

**Fig 4. Comparison of FRS-R progression rate from enrollment to death among subgroups.**

outcome [8]. Some researchers speculate that the shortening of survival may be related to decreased compliance with treatment methods such as non-invasive ventilation (NIV), which, according to clear empirical evidence, prolongs the survival time of ALS patients [19–21].

All three ALS-bvFTD patients in this study had bulbar onset. Although the association between bulbar symptoms and ALS-FTD has been a concern recognized by researchers, whether bulbar onset or bulbar involvement is correlated with ALS-FTD has not been determined [22]. Further evidence has shown that ALS-FTD is associated with bulbar involvement and that bulbar involvement is related to cognitive scores [10, 11]. Therefore, whether the

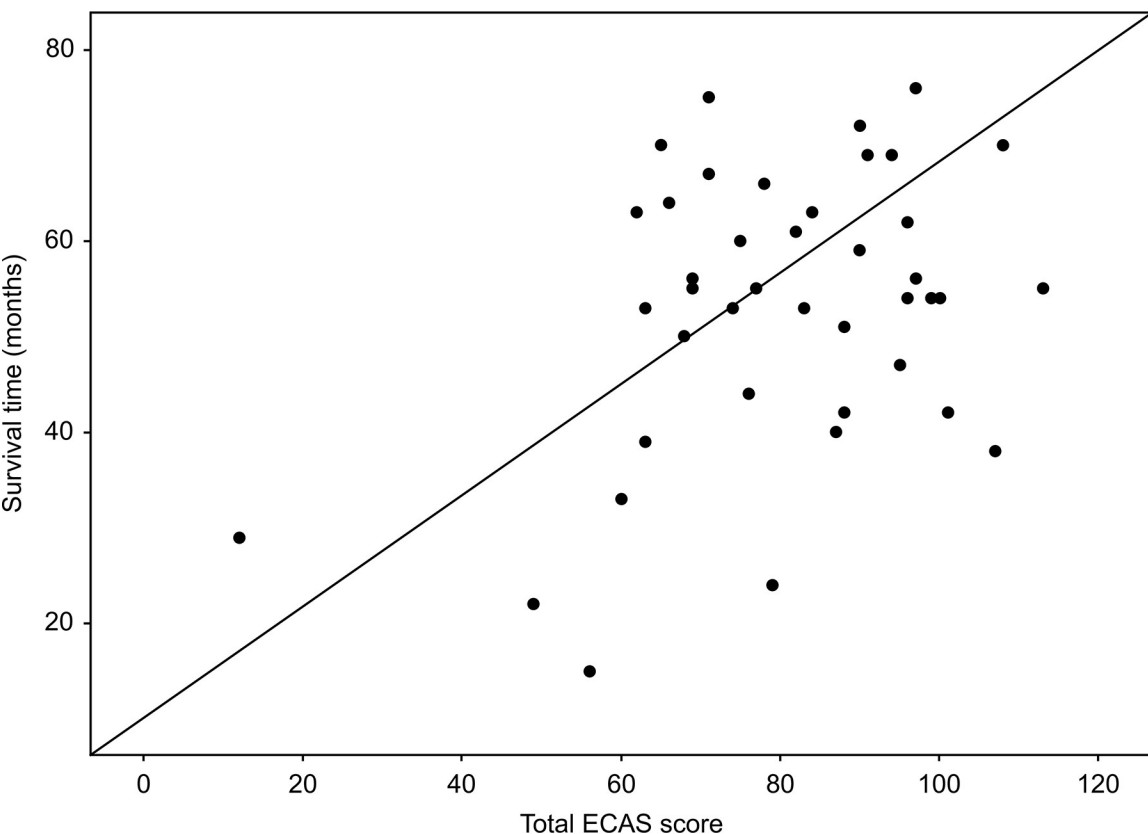

**Fig 5. Correlation between total ECAS score and survival time.**

shortened survival time of ALS-bvFTD is caused by frontotemporal dysfunction itself, behavioural disorder that impacts treatment, or bulbar involvement (which is thought to correlate with shorter survival time) is not known; a larger sample size is needed to explore the causes by multiple factor analysis.

In this study, there was no significant difference in the survival time between the pure ALS patients and ALS with cognitive and behavioural changes group that did not meet the diagnostic criteria of ALS-FTD [2]. The proportion of ALS-FTD is relatively small, while the percentage of mild cognitive and behavioural impairment is high. Therefore, this group of ALS patients is also worthy of attention. In our results, the survival time of the ALS-bvFTD group was significantly shorter than that of the ALS with cognitive and behavioural change group, which was consistent with the results of the correlation between the total ECAS score and survival time. However, previous studies have found that the rate of decline in cognitive and behavioural function in ALS is heterogeneous [9, 14, 23]. Our results also show that the FRS-R progress rate from enrolment to endpoint of ALS-bvFTD patients is significantly increased, and there is no significant correlation between cognitive score and ALS-FRS-R score, which reflects motor function changes. We speculated that the survival time of the ALS with cognitive and behavioural changes group was not shorter than that of the pure ALS group, possibly because the cognitive decline of this group of patients was slow. Motor dysfunction declines more rapidly and leads to death than cognitive deterioration, which might also shorten the survival time. However, this speculation needs to be further confirmed by follow-up studies.

The rate of loss to follow-up was 26%, which was similar with previous studies. However, the rate of follow-up to the endpoint was low, which may be related to the long follow-up time.

The main reason for the loss to follow-up was that the family members refused to continue follow-up or could not be contacted. Woolley et al. observed that during the one-year follow-up, the proportion of ALS patients without behavioural involvement was significantly higher than that of ALS patients with behavioural involvement [9]. In contrast, in this study, the proportion of patients lost to follow-up in the ALS-bvFTD group was the lowest (25.00%), followed by the ALS with cognitive and behavioural changes (48.78%) and pure ALS patient groups (57.14%). We speculate that the large difference may be related to the follow-up object and the follow-up time. Woolley et al. followed up the patients themselves, and after only one year of follow-up, patients with normal behaviour might have higher compliance. However, as time passes, the proportion of patients lost to follow-up increases, and the influence of behavioural factors may be continuously weakened. This study included 60 months of follow-up. The endpoint events were mainly obtained by asking the family members of patients. Therefore, the impact of cognitive and behavioural impairment on the results was relatively small. In addition, due to the significantly shortened survival time of ALS-bvFTD patients, the proportion of patients' families who could cooperate to complete the follow-up may be higher.

This is a 60-month long-term follow-up study with death and tracheotomy as an endpoint. However, the study design does not enable cognitive assessment of the endpoint; this outcome is mainly described by the family members, who are easily lost to follow-up. The participant of this study were mainly based on the published article in 2016 [15], and followed up for 60 months. The ALS-bvFTD diagnosis can be made clearly according to the criteria. However, if only use ECAS to evaluate, it can't fulfil with Strong's criteria of three intermediate groups (ALS with cognitive impairment, ALSci; ALS with behavioural impairment, ALSbi; and ALS with cognitive and behavioural impairment, ALScbi). Since ECAS has better sensitivity than specificity, it's sensitive to recognize those "pure ALS" patients. So we defined those with abnormal ECAS score and/or behavioural symptoms but can't meet Strong's ALS-FTD criteria [2] as ALS with cognitive and behavioural change group. However, it might increase the false positive rate in ALS with cognitive and behavioural change group, and the classification needs better design in future study. One of the important limitations of this study was that the sample size for follow-up was small; for this reason, Cox regression analysis was hard to be conducted, although such analysis would have been useful for the comparison of survival time, FRS-R progression rate, and the correlation between total ECAS score and survival time. This needs to be tested in further larger sample size study. In addition, cognitive and behavioural assessments were not conducted in the middle of the follow-up; thus, the degree of cognitive and behavioural dysfunction could not be evaluated in this study and will need to be addressed in future studies.

## Conclusion

ALS-bvFTD patients have shorter survival time. The total ECAS score may be correlated with survival time.

## Supporting information

**S1 Checklist. STROBE checklist.**
(DOCX)

**S2 Checklist. Clinical studies checklist.**
(DOCX)

**S3 Checklist. TRIPOD checklist.**
(DOCX)

**S1 Fig. Minimal data set for Fig 1.**
(XLSX)

**S2 Fig. Minimal data set for Fig 2.**
(XLSX)

**S3 Fig. Minimal data set for Fig 3.**
(XLSX)

**S4 Fig. Minimal data set for Fig 4.**
(XLSX)

**S5 Fig. Minimal data set for Fig 5.**
(XLSX)

**S1 Table. Minimal data set for Table 1.**
(XLSX)

**S2 Table. Minimal data set for Table 2.**
(XLSX)

**S3 Table. Minimal data set for Table 3.**
(XLSX)

## Author Contributions

**Formal analysis:** Shan Ye.

**Investigation:** Shan Ye, Pingping Jin, Lu Chen, Nan Zhang.

**Methodology:** Shan Ye.

**Project administration:** Shan Ye.

**Supervision:** Dongsheng Fan.

**Writing – original draft:** Shan Ye.

**Writing – review & editing:** Dongsheng Fan.

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
