## [Decision Letter · Decision Letter 0]

15 Feb 2021

PONE-D-20-39850

Prognosis of amyotrophic lateral sclerosis with cognitive and behavioural changes based on a five-year longitudinal follow-up

PLOS ONE

Dear Dr. Fan,

Thank you for submitting your manuscript to PLOS ONE. After careful consideration, we feel that it has merit but does not fully meet PLOS ONE’s publication criteria as it currently stands. Therefore, we invite you to submit a revised version of the manuscript that addresses the points raised during the review process.

We look forward to receiving your revised manuscript.

Kind regards,

Weidong Le

Academic Editor

PLOS ONE

Journal Requirements:

2. Please describe in your methods section how capacity to provide consent was determined for the participants in this study. Please also state whether your ethics committee or IRB approved this consent procedure. If you did not assess capacity to consent please briefly outline why this was not necessary in this case.

"This work was partially supported by grants from National Natural Science Foundation of China

(Project No. 82001350), Peking University Third Hospital Key Clinical Projects (Project No.

BYSY2018048), Peking University Third Hospital Cohort Construction Project (Project No.

BYSYDL2019002)"

Reviewers' comments:

Reviewer's Responses to Questions

**Comments to the Author**

1. Is the manuscript technically sound, and do the data support the conclusions?

Reviewer #1: No

Reviewer #2: Partly

Reviewer #3: Yes

2. Has the statistical analysis been performed appropriately and rigorously? 

Reviewer #1: No

Reviewer #2: Yes

Reviewer #3: No

3. Have the authors made all data underlying the findings in their manuscript fully available?

Reviewer #1: No

Reviewer #2: Yes

Reviewer #3: No

4. Is the manuscript presented in an intelligible fashion and written in standard English?

Reviewer #1: No

Reviewer #2: Yes

Reviewer #3: Yes

5. Review Comments to the Author

Reviewer #1: This is an original article about the role of cognitive impairment as a prognostic determinant in a small and heterogeneous group of ALS patient. Despite the interesting topic, the article completely lacks data to support its conclusions. First, the definition of cognitive impairment did not follow correctly the mentioned Strong criteria for ALS-FTD; also, the author used only ECAS to define cognitive categories. Second, the statistical analysis for survival is not a survival analysis (Kaplan-Meier or Cox proportional HM) but a correlation analysis, performed only on deceased patients – a significant source of bias. Third, there is no statistical comparison of the three, small and heterogeneous groups and no adjustment for other prognostic determinant, such as age, sex, bulbar vs spinal onset. The article should not be published in the current form.

Abstract:

1. The ALS-FTD is ALS plus frontotemporal dementia and not only behavioral variant, which is the most common. Please rephrase.

2. Less than half of the patients were followed-up (42 from 87).

3. Survival time is in years?

Materials and methods

4. As in the abstract, check the definition for ALS-FTD (ALS with fronto-temporal dementia) using Strong Criteria.

5. In this section, the authors did not mention any details about the neuropsychological battery used to derive cognitive classification, did they use only the ECAS? Moreover to clarify this process, I think it is better to classify patients in ALL the Strong categories contained in the Axix II of the original article (ALS-CN, ALS-ci, ALS-bi, ALS-cbi, and ALS-FTD) and then, due to the relative small number of patients included, to merge intermediate cognitive impairment into a unique category. If only ECAS was used to define cognitive status, it is very difficult to believe in this classification.

6. ECAS score published previously: citation? I think it is better to reassume patients features also in this article, not only adding the clinical information of the 3 ALS-FTD patients added. For this, Table 1 add no useful information without comparison to the other patients included.

7. If the authors said that all patients were followed up, why the only a half of patients complete the follow up? Do you mean they died? Please clarify this point.

8. Substitute “clinical death” with “death”.

9. Statistics Pearson and Spearman correlation are not survival analysis. This is the major problem of the article, use this statistical methods, combined with a small number of patients, analyzing only data from patients who reach the outcome is a huge source of bias.

Results

10. Table 2. There is no comparison among groups… for example, the disease duration is very different among groups, especially in the CN-group the survival is significantly higher than in other groups.

11. The reduced survival could be related to other clinical features, more than cognitive impairment, such as age (62 vs 55), male sex, the higher percentage of bulbar-onset patients.. without Cox analysis the authors are not able to consider all this features in adjustments.

Conclusions

12. The conclusions continue to refer only to behavioral impairment, misinterpreting the Strong definitions for ALS-FTD. Considering the multiple possible bias performed in the Results section, they should be completely rewritten.

Reviewer #2: The authors of this paper have followed up for 5 years a cohort of 87 ALS patients classified according to cognitive impairment. ALS-FTD patients had a worse survival and more rapid progression of ALSFRS-R compared to ALS with cognitive and behaviour changes and cognitively normal ALS. The total Edinburgh Cognitive and Behavioural Amyotrophic Lateral Sclerosis Screen (ECAS) score was positively correlated with survival time. The authors conclude that patients with ALS-FTD have a shorter survival time than patients with simple ALS or ALS combined with cognitive and behavioural dysfunction.

Some comments

1. The classification of patients in ALS-FTD, ALS with cognitive and behaviour changes and ‘simple ALS’ is not clear. The authors refer to Strong’s classification, but this classification propose 3 intermediate groups (ALS with cognitive impairment, ALSci; ALS with behavioural impairment, ALSbi; and ALS with cognitive and behavioural impairment, ALScbi). It seems that the authors have reunited the 3 intermediate forms. If it is correct, they should explain this point in the methods section.

2. A second relevant point: was the cognitive classification based only on ECAS or the authors used a more comprehensive battery of cognitive tests? This aspect should be better explained. However, if only ECAS was used, this should be reported as a limitation of the study, since ECAS alone cannot allow to categorize the patients according to Strong’s classification.

3. An interesting finding of the paper is the correlation between ECAS and survival. It would be interesting for the readers to know if only the total ECAS score was corrected with survival or if any of the subscores were also correlated.

4. Among paper suggesting that cognitive impairment worsens over time the authors missed the following one, which should be included in the discussion since is demonstrates that patients who have a worsening of their cognitive function has a also a worse outcome.

Bersano E, Sarnelli MF, Solara V, Iazzolino B, Peotta L, De Marchi F, Facchin AP, Moglia C, Canosa A, Calvo A, Chiò A, Mazzini L. Decline of cognitive and behavioral functions in amyotrophic lateral sclerosis: a longitudinal study. Amyotroph Lateral Scler Frontotemporal Degener. 2020 Aug;21(5-6):373-379. doi: 10.1080/21678421.2020.1771732.

5. The reasons for patients’ drop-out/loss to follow-up should be reported.

Minor point

Citation 3 is wrong. Iazzolino is mispelled as Lazzolino

Reviewer #3: The authors provided a prognostic study in patients with ALS to explore the cognitive and behavior changes in a longitudinal study. The study was of importance to explore the interaction between ALS progression and cognitive and behavior changes. However, there were several critiques need to be further addressed.

1. In the method section, the diagnostic level of ALS patients who met the revised El Escorial diagnostic criteria enrolled in this study need to be clarified.

2. In the method section, patients with severe physical dysfunction were excluded in this study. However, ALS is a progressive disease and more than half of patients were loss to follow-up. Therefore, the baseline characteristics of those failed to follow-up need to be clarified and it is essential to compare those patients with follow-up and no follow-up and to tell if the assumptions were made about verification (ignorable or missing at random–MAR) or non-ignorable or missing not at random–MNAR)

3. In the method section, the authors stated” The clinical information and neuropsychological Edinburgh Cognitive and Behavioural Amyotrophic Lateral Sclerosis Screen (ECAS) scores of 84 of these ALS patients have been published previously.” However, there was no reference cited. In addition, what is the relationship between those to the other three in Table 1?

4. In Table1, diagnostic levels statement was better to be detailed, ALS level or ALS-FTD level ?

5. In the method section, the ECAS was a brief multidomain assessment originally designed for people with Amyotrophic Lateral Sclerosis (ALS/Motor Neurone Disease) but is also useful in other neurodegenerative disorders. However, it was not the golden standard of FTD diagnosis. The criteria of ALS-FTD may need to be further clarified.

6. In statistical section, ALS-FRS was the outcome, and ECAS was the variates. Both were dynamic changed during the follow-up. However, FRS-R progression rate was not supposed as a consistent slope. Therefore, the Cox proportional regression model should be used. Also, modification effect and interaction between onset of disease need be further demonstrated. All statistical strategy need to be re-analyzed and confirmed.

7. In result section, the authors stated that there were differences of total ECAS score between groups. However, it looks similar to the definition of grouping. The meaning of different level of ECAS score means their varied level of cognitive performance, which was confusing. The goal of study need to be further addressed.

8. In the result section, prototypical STARD diagram to report flow of participants through the study was need in the results

9. TRIPOD checklist for prediction model development is highly recommended to be used in this study and uploaded as supplemental materials.

This would be an interesting study and identified the relationship between ALS and their cognitive and behavioral changes

6. PLOS authors have the option to publish the peer review history of their article (what does this mean?). If published, this will include your full peer review and any attached files.

Reviewer #1: No

Reviewer #2: No

Reviewer #3: No

---

## [Author Response · Author response to Decision Letter 0]

11 Mar 2021

Reviewer #1: This is an original article about the role of cognitive impairment as a prognostic determinant in a small and heterogeneous group of ALS patient. Despite the interesting topic, the article completely lacks data to support its conclusions. First, the definition of cognitive impairment did not follow correctly the mentioned Strong criteria for ALS-FTD; also, the author used only ECAS to define cognitive categories. Second, the statistical analysis for survival is not a survival analysis (Kaplan-Meier or Cox proportional HM) but a correlation analysis, performed only on deceased patients – a significant source of bias. Third, there is no statistical comparison of the three, small and heterogeneous groups and no adjustment for other prognostic determinant, such as age, sex, bulbar vs spinal onset. The article should not be published in the current form.

 Reply: Thank you very much for your comments. They are very helpful for revising the manuscript. We have revised them according to your advices, and the details of the reply for each comment above are seperately written under the following questions.

Abstract:

1. The ALS-FTD is ALS plusfrontotemporal dementia and not only behavioral variant, which is the most common. Please rephrase.

Reply: Thank you for the advice. It’s easy to make confusion if use ALS-FTD. Since this group of patients were all bvFTD, we rephrase this group as ALS-bvFTD group.

2. Less than half of the patients were followed-up (42 from 87).

Reply: Thank you for the question. This question and your question 7 and question 9 were very helpful to revise the manuscript. Half of the “loss to follow-up patients” were not completely loss, but took the early follow-up. However, we couldn’t contact them later before record the endpoint event. We reply this question in detail in question 7 and 9. Thanks again.

3. Survival time is in years?

Reply: Thank you for you reminder. We have change “All patients were followed up for 5 years” into “All patients were followed up for 60 months”. The other part in the manuscript has also been changed.

Materials and methods

4. As in the abstract, check the definition for ALS-FTD (ALS with fronto-temporal dementia) using Strong Criteria.

 Reply: Thanks for the advice. As in the abstract, all the patients in this group fulfill the ALS-bvFTD critiria. In order to avoid confusion, we rephrase this group as ALS-bvFTD group. It has already been changed in the manuscript.

5. In this section, the authors did not mention any details about the neuropsychological battery used to derive cognitive classification, did they use only the ECAS? Moreover to clarify this process, I think it is better to classify patients in ALL the Strong categories contained in the Axix II of the original article (ALS-CN, ALS-ci, ALS-bi, ALS-cbi, and ALS-FTD) and then, due to the relative small number of patients included, to merge intermediate cognitive impairment into a unique category. If only ECAS was used to define cognitive status, it is very difficult to believe in this classification.

Reply: Thank you very much for the advices. The study participants were mainly based on the published article (validation of ECAS Chinese version) in 2016, and followed up for 5 years. We rewrote this paragraph, as following:” The clinical information and neuropsychological Edinburgh Cognitive and Behavioural Amyotrophic Lateral Sclerosis Screen (ECAS) scores of 84 of these ALS patients have been published previously15.As reported in this article, one patient met ALS-bvFTD criteria according to Rascovsky17 and Strong criteria2. Forty-two patients with normal ECAS score and no behavioural symptom were defined as pure ALS group. The other forty-one patients with abnormal ECAS score and/or behavioural symptoms but can’t fulfill Strong’s ALS-FTD criteria2 were defined as ALS with cognitive and behavioural group. There were three ALS-bvFTD patients enrolled during the same period. The three patients together with the one reported, were defined as ALS-bvFTD group. The clinical information of 87 ALS patients were shown in Table 1.”

In the 2016 paper, we only used ECAS to divide cognitive and behavioural impairment. This was an important limitation as you mentioned, and also the reason we didn’t divided the three intermediate groups (ALSci, ALSbi, ALScbi). So we add a paragraph for limitation in the discussion part, as following:” The participant of this study were mainly based on the published article in 201615, and followed up for 5 years. The ALS-bvFTD diagnosis can be made clearly according to the criteria. However, if only use ECAS to evaluate, it can’t fulfill with Strong’s criteria of three intermediate groups (ALS with cognitive impairment, ALSci; ALS with behavioural impairment, ALSbi; and ALS with cognitive and behavioural impairment, ALScbi). Since ECAS has better sensitivity than specificity, it’s sensitive to recognize those “pure ALS” patients. So we defined those with abnormal ECAS score and/or behavioural symptoms but can’t meet Strong’s ALS-FTD criteria2 as ALS with cognitive and behavioural change group. However, it might increase the false positive rate in ALS with cognitive and behavioural change group, and the classification needs better design in future study.”

 Thanks again for your comments.

6. ECAS scorepublished previously: citation? I think it is better to reassume patients features also in this article, not only adding the clinical information of the 3 ALS-FTD patients added. For this, Table 1 add no useful information without comparison to the other patients included.

Reply: Sorry that the citation number (No.15) was mistakenly deleted. It has been added. Thanks for your kind advice. We deleted original Table 1, and list the information of all 87 patients as new Table1. The details were listed in the manuscript.

7. If the authors said that all patients were followed up, why the only a half of patients complete the follow up? Do you mean they died? Please clarify this point.

Reply: Thank you very much for the question. In the original manuscript, only a half of patients completed the follow up. The reason was that among the 50% “loss to follow-up patients”, half of them were not completely loss, but took the early follow-up. However, we couldn’t contact them later before record the endpoint event. Thanks to this question and question 9, they are very helpful for revising the manuscript. So this time, we put this part of patients and those who reached the endpoint together to do K-M survival analysis, and get a similar result.

In the manuscript, we add a paragraph to descript this in Material part, as following: “All patients were followed up for 5 years, during which telephone follow-up was given every half year, until loss to follow-up or getting the end point. The main end point was death and tracheotomy. During the 5 years, 42 of 87 ALS patients reached the endpoint, among which 3 patients were taken tracheotomy, 27 patients died of respiratory failure, 1 patient died of multiple organ failure, and the cause of 11 patients were unknown; 22 ALS patients received early follow-up, but refused further follow-up or could not be contacted before the endpoint; 23 ALS patients (26.44%) were loss to follow-up. ”

8. Substitute “clinical death” with “death”.

Reply: Thank you very much. We have substituted.

9. Statistics Pearson and Spearman correlation are not survival analysis. This is the major problem of the article, use this statistical methods, combined with a small number of patients, analyzing only data from patients who reach the outcome is a huge source of bias.

 Reply: Thank you for your important comment. Combining with the reply in question7, we didn’t contain the patients who failed to follow up to the endpoint and used correlation analysis. These would bring bias. So according to your advices, we included this part of patients to do K-M survival analysis, and added this as part 1 in the results, as following:” 1. Survival analysis of ALS patients with and without cognitive and behavioural changes When taking the Kaplan-Meier survival analysis, the survival curve of pure ALS group and ALS with cognitive and behavioural change group had no difference, while the survival time of ALS-bvFTD group was significant shorter (Figure 2).” Thanks again for your advices.

Results

10. Table 2. There is no comparison among groups… for example, the disease duration is very different among groups, especially in the CN-group the survival issignificantly higher than in other groups.

 Reply: Thank you very much for your question. Since there are three groups, comparison between each of the two groups will make the table seem to be complicated. So we described the comparison results in words (Manuscript: Result—2.1 Clinical information of ALS patients followed up to the endpoint). About the survival time, since it’s an important prognosis factor, we put it together with FRS-R progression rate as a separate part with figures(Manuscript: Result—2.2 The characteristics of progression prognosis in different subgroups). This was also the reason that didn’t put comparison results in Table 2. Thanks again.

11. The reduced survival could be related to other clinical features, more than cognitive impairment, such as age (62 vs 55), male sex, the higher percentage of bulbar-onset patients.. without Cox analysis the authors are not able to consider all this features in adjustments.

Reply: Thank a lot for your comment. We quite agree with you. So we also discuss this in the discussion part, as “All three ALS-bvFTD patients in this study had bulbar onset. Although the association between bulbar symptoms and ALS-FTD has been a concern recognized by researchers, whether bulbar onset or bulbar involvement is correlated with ALS-FTD has not been determined22. Further evidence has shown that ALS-FTD is associated with bulbar involvement and that bulbar involvement is related to cognitive scores 10,11. Therefore, whether the shortened survival time of ALS-FTD is caused by frontotemporal dysfunction itself, behavioural disorder that impacts treatment, or bulbar involvement (which is thought to correlate with shorter survival time) is not known; a larger sample size is needed to explore the causes by multiple factor analysis.”

Also we agree with you that it needs cox analysis to get rid of the other factors, such as age, education et al. However, the sample size is limited to do this, especially ALS-bvFTD group. So we wrote this limitation in the discussion part, as following: “One of the important limitations of this study was that the sample size for follow-up was small; for this reason, Cox regression analysis was hard to be conducted, although such analysis would have been useful for the comparison of survival time, FRS-R progression rate, and the correlation between total ECAS score and survival time. This needs to be tested in further larger sample size study.” Thanks again for your question.

Conclusions

12. The conclusions continue to refer only to behavioral impairment, misinterpreting the Strong definitions for ALS-FTD. Considering the multiple possible bias performed in the Results section, they should be completely rewritten.

 Reply: Thank you for your advices all above. We had revised carefully according to your advices. The method and statistics has been revised. It’s happy to find, after revised, ALS-bvFTD patients still seem to have worse prognosis. We quite agree with you that there are limitations of this study. Since the patients were based on the published study, some basic information couldn’t be changed. However, it’s a 5-year long time follow-up, and can get patients death endpoint. We are wondering it might be a preliminary exploration for further large sample and well-designed prospective cohort study. Thanks indeed for all of your comment again, they are helpful for revised the study.

Reviewer #2: The authors of this paper have followed up for 5 years a cohort of 87 ALS patients classified according to cognitive impairment. ALS-FTD patients had a worse survival and more rapid progression of ALSFRS-R compared to ALS with cognitive and behaviour changes and cognitively normal ALS. The total Edinburgh Cognitive and Behavioural Amyotrophic Lateral Sclerosis Screen (ECAS)score was positively correlated with survival time. The authors conclude that patients with ALS-FTD have a shorter survival time than patients with simple ALS or ALS combined with cognitive and behavioural dysfunction.

Reply: Thank you very much for your comments.

Some comments

1. The classification of patients in ALS-FTD, ALS with cognitive and behaviour changes and ‘simple ALS’ is not clear. The authors refer to Strong’s classification, but this classification propose 3 intermediate groups (ALS with cognitive impairment, ALSci; ALS with behavioural impairment, ALSbi; and ALS with cognitive and behavioural impairment, ALScbi). It seems that the authors have reunited the 3 intermediate forms. If it is correct, they should explain this point in the methods section.

Reply: Thanks a lot for your advices. The classification of the 3 groups were not described clearly. So we add this paragraph in “Materials--Participants” part, as following: “The clinical information and neuropsychological Edinburgh Cognitive and Behavioural Amyotrophic Lateral Sclerosis Screen (ECAS) scores of 84 of these ALS patients have been published previously15.As reported in this article, one patient met ALS-bvFTD criteria according to Rascovsky17 and Strong criteria2. Forty-two patients with normal ECAS score and no behavioural symptom were defined as pure ALS group. The other forty-one patients with abnormal ECAS score and/or behavioural symptoms but can’t fulfill Strong’s ALS-FTD criteria2 were defined as ALS with cognitive and behavioural change group. There were three ALS-bvFTD patients enrolled during the same period. The three patients together with the one reported, were defined as ALS-bvFTD group. The clinical information of 87 ALS patients was shown in Table 1.”

2. A second relevant point: was the cognitive classification based only on ECAS or the authors used a more comprehensive battery of cognitive tests? This aspect should be better explained. However, if only ECAS was used,this should be reported as a limitation of the study, since ECAS alone cannot allow to categorize the patients according to Strong’s classification.

Reply: Thank you very much. We quite agree with your advice. The study participants were mainly based on the published article (validation of ECAS Chinese version) in 2016, and followed up for 5 years. So in the 2016 paper, we only used ECAS to divide cognitive and behavioural impairment. This was an important limitation as you mentioned, and also the reason we didn’t divided the three intermediate groups (ALSci, ALSbi, ALScbi). So we add a paragraph in the discussion part, as following:” The participant of this study were mainly based on the published article in 201615, and followed up for 5 years. The ALS-bvFTD diagnosis can be made clearly according to the criteria. However, if only use ECAS to evaluate, it can’t fulfill with Strong’s criteria of three intermediate groups (ALS with cognitive impairment, ALSci; ALS with behavioural impairment, ALSbi; and ALS with cognitive and behavioural impairment, ALScbi). Since ECAS has better sensitivity than specificity, it’s sensitive to recognize those “pure ALS” patients. So we defined those with abnormal ECAS score and/or behavioural symptoms but can’t meet Strong’s ALS-FTD criteria2 as ALS with cognitive and behavioural change group. However, it might increase the false positive rate in ALS with cognitive and behavioural change group, and the classification needs better design in future study.”

3. An interesting finding of the paper is the correlation between ECAS and survival. It would be interesting for the readers to know if only the total ECAS score was corrected with survival or if any of the subscores were also correlated.

Reply: Thanks a lot for your advice. We had added this in the result part, as following:” As for the ECAS subscores, language (r = 0.35, p = 0.02) and ALS-specific function score (r = 0.42, p = 0.01) had positive correlation with survival time, while the others didn’t (p > 0.05).” While as the author of ECAS validated, ECAS total score as a whole has best specificity and sensitivity, ECAS total score may be used more frequently. However, it’s still interesting to explore the subfuntions. Thanks again for your advice.

4. Among paper suggesting that cognitive impairment worsens over time the authors missed the following one, which should be included in the discussion since is demonstrates that patients who have a worsening of their cognitive function has a also a worse outcome.

Bersano E, Sarnelli MF, Solara V, Iazzolino B, Peotta L, De Marchi F, Facchin AP, Moglia C, Canosa A, Calvo A, Chiò A, Mazzini L. Decline of cognitive and behavioral functions in amyotrophic lateral sclerosis: a longitudinal study. Amyotroph Lateral Scler Frontotemporal Degener. 2020 Aug;21(5-6):373-379.doi: 10.1080/21678421.2020.1771732.

Reply: Thank you very much for your kind reminder. It’s a very important study. We had added it separately in the background and discussion. ALS-FTD patients usually had worsening of cognitive function and had shorter survival time. It’s very interesting to pay more attention to the deterioration of cognition for prognosis prediction. Thanks again.

5. The reasons for patients’ drop-out/loss to follow-up should be reported.

Reply: Thank you for this question. Half of the “loss to follow-up patients” were not completely loss, but took the early follow-up. However, we couldn’t contact them later before record the endpoint event. We didn’t contain this part of patient in the original manuscript, it might bring bias. So this time, we put this part of patients and those who reached the endpoint together to do K-M survival analysis, and get a similar result.

In the manuscript, we add a paragraph to descript this in Material part, as following: “All patients were followed up for 5 years, during which telephone follow-up was given every half year, until loss to follow-up or getting the end point. The main end point was death and tracheotomy. During the 5 years, 42 of 87 ALS patients reached the endpoint, among which 3 patients were taken tracheotomy, 27 patients died of respiratory failure, 1 patient died of multiple organ failure, and the cause of 11 patients were unknown; 22 ALS patients received early follow-up, but refused further follow-up or could not be contacted before the endpoint; 23 ALS patients (26.44%) were loss to follow-up. ”

Minor point

Citation 3 is wrong. Iazzolino is mispelled as Lazzolino

Reply: Thank a lot for your kind reminder. It has been revised in the references part.

Reviewer #3: The authors provided a prognostic study in patients with ALS to explore the cognitive and behavior changes in a longitudinal study. The study was of importance to explore the interaction between ALS progression and cognitive and behavior changes. However, there were several critiques need to be further addressed.

Reply: Thank you very much for your kind and helpful advices. We have revised the manuscript according to your comments.

1. In the method section, the diagnostic level of ALS patients who met the revised El Escorial diagnostic criteria enrolled in this study need to be clarified.

 Reply: Thank you for your advice. We had revised Table 1 in the manuscript to list clinical information of all 87 patients, including diagnostic level. The detail is in “Materials and Methods—1.Participants—Table1”.

2. In the method section, patients with severe physical dysfunction were excluded in this study. However, ALS is a progressive disease and more than half of patients were loss to follow-up. Therefore, the baseline characteristics of those failed to follow-up need to beclarified and it is essential to compare those patients with follow-up and no follow-up and to tell if the assumptions were made about verification (ignorable or missing at random–MAR) or non-ignorable or missing not at random–MNAR)

Reply: Thanks a lot for your advices, they are important and helpful. Firstly, about the excluding of severe physical dysfunction patients, the patients in this study was mainly based on the published study in 2016. (Ye S, et al. The Edinburgh Cognitive and Behavioural ALS Screen in a Chinese Amyotrophic Lateral Sclerosis Population[J]. PLoS One. 2016;11(5):e0155496.) In this study, severe physical dysfunction patients were excluded in order to make sure patients were able to finish ECAS examination. 

Then, about half of patients were loss to follow-up. According to your advice, we looked back our data carefully. Half of the “loss to follow-up patients” were not completely loss, but took the early follow-up. However, we couldn’t contact them later before record the endpoint event. We didn’t contain this part of patient in the original manuscript, it might bring bias as you mentioned. So this time, we put this part of patients and those who reached the endpoint together to do K-M survival analysis. It’s happy to find, they get similar results.

In the manuscript, we add a paragraph to descript this in Material part, as following: “All patients were followed up for 5 years, during which telephone follow-up was given every half year, until loss to follow-up or getting the end point. The main end point was death and tracheotomy. During the 5 years, 42 of 87 ALS patients reached the endpoint, among which 3 patients were taken tracheotomy, 27 patients died of respiratory failure, 1 patient died of multiple organ failure, and the cause of 11 patients were unknown; 22 ALS patients received early follow-up, but refused further follow-up or could not be contacted before the endpoint; 23 ALS patients (26.44%) were loss to follow-up. ”

3. In the method section, the authors stated” The clinical information and neuropsychological Edinburgh Cognitive and Behavioural Amyotrophic Lateral Sclerosis Screen (ECAS) scores of 84 of these ALS patients have been published previously.” However, there was no reference cited. In addition, what is the relationship between those to the other three in Table 1?

Reply: Sorry that the citation number (No.15) was mistakenly deleted. It has been added. (Ye S, et al. The Edinburgh Cognitive and Behavioural ALS Screen in a Chinese Amyotrophic Lateral Sclerosis Population[J]. PLoS One. 2016;11(5):e0155496.) Thanks for your kind reminder. 

In the ECAS study in 2016, there were another 3 ALS-bvFTD patients that were not included in. So we add them together for the follow-up study. We rewrote this paragraph to make it clearly, as following:” The clinical information and neuropsychological Edinburgh Cognitive and Behavioural Amyotrophic Lateral Sclerosis Screen (ECAS) scores of 84 of these ALS patients have been published previously15.As reported in this article, one patient met ALS-bvFTD criteria according to Rascovsky17 and Strong criteria2. Forty-two patients with normal ECAS score and no behavioural symptom were defined as pure ALS group. The other forty-one patients with abnormal ECAS score and/or behavioural symptoms but can’t fulfill Strong’s ALS-FTD criteria2 were defined as ALS with cognitive and behavioural change group. There were three ALS-bvFTD patients enrolled during the same period. The three patients together with the one reported, were defined as ALS-bvFTD group. The clinical information of 87 ALS patients was shown in Table 1.”

We deleted original Table 1, and list the information of all 87 patients as new Table1. The details were listed in the manuscript.

4. In Table1, diagnostic levels statement was better to be detailed, ALS level or ALS-FTD level ?

Reply: Thank you for your kind advice. The new Table 1 has contained the information. The details were listed in the manuscript: “Materials and Methods—1.Participants—Table1”..

5. In the method section, the ECAS was a brief multidomain assessment originally designed for people with Amyotrophic Lateral Sclerosis (ALS/Motor Neurone Disease) but is also useful in other neurodegenerative disorders. However, it was not the goldenstandard of FTD diagnosis. The criteria of ALS-FTD may need to be further clarified.

Reply: Thank you very much for your comments. We quite agree with you. The study participants were mainly based on the published article (validation of ECAS Chinese version) in 2016, and followed up for 5 years. So in the 2016 paper, we only used ECAS to divide cognitive and behavioural impairment. This was an important limitation as you mentioned, and also the reason we didn’t divided the three intermediate groups (ALSci, ALSbi, ALScbi). So we add a paragraph to discuss the limitation, as following:” The participant of this study were mainly based on the published article in 201615, and followed up for 5 years. The ALS-bvFTD diagnosis can be made clearly according to the criteria. However, if only use ECAS to evaluate, it can’t fulfill with Strong’s criteria of three intermediate groups (ALS with cognitive impairment, ALSci; ALS with behavioural impairment, ALSbi; and ALS with cognitive and behavioural impairment, ALScbi). Since ECAS has better sensitivity than specificity, it’s sensitive to recognize those “pure ALS” patients. So we defined those with abnormal ECAS score and/or behavioural symptoms but can’t meet Strong’s ALS-FTD criteria2 as ALS with cognitive and behavioural change group. However, it might increase the false positive rate in ALS with cognitive and behavioural change group, and the classification needs better design in future study.”

6. In statistical section, ALS-FRS was the outcome, and ECAS was the variates. Both were dynamic changed during the follow-up. However, FRS-R progression rate was not supposed as a consistent slope. Therefore, the Cox proportional regression model should be used. Also, modification effect and interaction between onset of disease need be further demonstrated. All statistical strategy need to be re-analyzed and confirmed.

Reply: Thank you indeed for your advices, we agree with you very much. There are several interaction factors, such as age, gender, onset of disease, diagnostic level, et al. For example, Onset of disease, as you mentioned, we discussed it especially in the discussion part:” All three ALS-bvFTD patients in this study had bulbar onset. Although the association between bulbar symptoms and ALS-FTD has been a concern recognized by researchers, whether bulbar onset or bulbar involvement is correlated with ALS-FTD has not been determined22. Further evidence has shown that ALS-FTD is associated with bulbar involvement and that bulbar involvement is related to cognitive scores 10,11. Therefore, whether the shortened survival time of ALS-FTD is caused by frontotemporal dysfunction itself, behavioural disorder that impacts treatment, or bulbar involvement (which is thought to correlate with shorter survival time) is not known; a larger sample size is needed to explore the causes by multiple factor analysis.”

Also we agree with you that it needs cox analysis to get rid of these factors. However, the sample size is limited, especially ALS-bvFTD group. So we wrote this limitation in the discussion part, as following: “One of the important limitations of this study was that the sample size for follow-up was small; for this reason, Cox regression analysis was hard to be conducted, although such analysis would have been useful for the comparison of survival time, FRS-R progression rate, and the correlation between total ECAS score and survival time. This needs to be tested in further larger sample size study”. Thanks again for your question.

7. In result section, the authors stated that there were differences of total ECAS score between groups. However, it looks similar to the definition of grouping. The meaning of different level of ECAS score means their varied level of cognitive performance, which was confusing. The goal of study need to be further addressed.

 Reply: Thank you for your kind reminder. This part was hoping to show the clinical characteristics of the patients who was followed up the endpoint. So the clinical information was list in Table 2, and to see whether there was any difference between three groups. Since it’s easy to make confusion of this part and the total patients, we changed the title of this part as “2.1 Clinical information of ALS patients followed up to the endpoint”

8. In the result section, prototypical STARD diagram to report flow of participants through the study was need in the results

 Reply: Thanks a lot for your advice. We have added the diagram as Figure 1. The details were listed in the manuscript.

9.TRIPOD checklist for prediction model development is highly recommended to be used in this study and uploaded as supplemental materials.

Reply: Thanks for your advice. It has been uploaded as supplemental materials.

This would be an interesting study and identified the relationship between ALS and their cognitive and behavioral changes

Reply: Thanks indeed for all above advices.

---

## [Decision Letter · Decision Letter 1]

19 Apr 2021

PONE-D-20-39850R1

Prognosis of amyotrophic lateral sclerosis with cognitive and behavioural changes based on a sixty-month longitudinal follow-up

PLOS ONE

Dear Dr. Fan,

Thank you for submitting your manuscript to PLOS ONE. After careful consideration, we feel that it has merit but does not fully meet PLOS ONE’s publication criteria as it currently stands. Therefore, we invite you to submit a revised version of the manuscript that addresses the points raised during the review process.

We look forward to receiving your revised manuscript.

Kind regards,

Weidong Le

Academic Editor

PLOS ONE

Journal Requirements:

Reviewers' comments:

Reviewer's Responses to Questions

**Comments to the Author**

1. If the authors have adequately addressed your comments raised in a previous round of review and you feel that this manuscript is now acceptable for publication, you may indicate that here to bypass the “Comments to the Author” section, enter your conflict of interest statement in the “Confidential to Editor” section, and submit your "Accept" recommendation.

Reviewer #2: All comments have been addressed

Reviewer #3: (No Response)

2. Is the manuscript technically sound, and do the data support the conclusions?

Reviewer #2: Yes

Reviewer #3: Yes

3. Has the statistical analysis been performed appropriately and rigorously? 

Reviewer #2: Yes

Reviewer #3: No

4. Have the authors made all data underlying the findings in their manuscript fully available?

Reviewer #2: Yes

Reviewer #3: Yes

5. Is the manuscript presented in an intelligible fashion and written in standard English?

Reviewer #2: Yes

Reviewer #3: Yes

6. Review Comments to the Author

Reviewer #2: All my comments were adequately addressed. I want to compliments the authors for this important and well written paper.

Reviewer #3: The authors revised the manuscript and responded all reviewer comments point to point. Still, I had some comment which had not been clarified. The major concerns were still the methodology and statistical analysis.

1. In the method section, patients with severe physical dysfunction were excluded in this study. The high rate of missing data was reasonable. However, it is essential to compare those patients with follow-up and no follow-up and to tell if the assumptions were made about verification (ignorable or missing at random–MAR) or non-ignorable or missing not at random–MNAR). The authors responsed “According to your advice, we looked back our data carefully. We didn’t contain this part of patient in the original manuscript, it might bring bias as you mentioned. ” No details about those patients was listed yet. The authors need to consider a column added to table 1 and table 2.

2. Since the missing data was unavoidable, the authors need to address censoring issue and modified their statistical strategy.

3. In the results section, the authors stated that survival time of the ALS-bvFTD group was significantly shorter than that of the pure ALS group. However, since the distribution was likely not normal, the Wilcoxon test or other approaches could be used.

4. The difference between pure ALS and cognitive impairment ALS was small. However, several baseline characteristics were needed to adjusted and log-rank test was need to be considered here.

5. The linear regression analysis between progression rate and follow-up time was performed. However，it might be nonlinear regression model and cubic or GEE model were worth be explored.

7. PLOS authors have the option to publish the peer review history of their article (what does this mean?). If published, this will include your full peer review and any attached files.

Reviewer #2: No

Reviewer #3: No

---

## [Author Response · Author response to Decision Letter 1]

28 May 2021

The authors revised the manuscript and responded all reviewer comments point to point. Still, I had some comment which had not been clarified. The major concerns were still the methodology and statistical analysis.

1. In the method section, patients with severe physical dysfunction were excluded in this study. The high rate of missing data was reasonable. However, it is essential to compare those patients with follow-up and no follow-up and to tell if the assumptions were made about verification (ignorable or missing at random–MAR) or non-ignorable or missing not at random–MNAR). The authors responsed “According to your advice, we looked back our data carefully. We didn’t contain this part of patient in the original manuscript, it might bring bias as you mentioned. ” No details about those patients was listed yet. The authors need to consider a column added to table 1 and table 2.

Reply: Thank you very much for your constructive suggestion. According to your advice, before the results part “Analysis of ALS patients followed up to the endpoint”, we added another part to compare ALS patients who followed up to the endpoint and not. We found except for ECAS score, actually there was no difference in other clinical factors. Since the percentage of followed up to the endpoint patients was much higher in ALS-bvFTD (3/4), and ECAS score of ALS-bvFTD patients are usually lower, we get rid of 4 ALS-bvFTD patients. Then there was also no difference of ECAS score between these two groups. According to these results, we then analyze ALS followed up to the endpoint group. The added result part was listed as following:

“2. Comparison between followed up to the endpoint and not to the endpoint groups

 Except for ECAS score, there was no significant difference between the ALS patients who followed up to the endpoint and not to the endpoint (Table2). The percentage of followed up to the endpoint patients was much higher in ALS-bvFTD (3/4), and ECAS score of ALS-bvFTD patients are usually lower. So when we got rid of the ALS-bvFTD patients, there was no difference of ECAS score between the two groups (t=1.55, p=0.12).

Table2. Comparison between followed up to the endpoint and not to the endpoint groups“

　 

2. Since the missing data was unavoidable, the authors need to address censoring issue and modified their statistical strategy.

Reply: Thank you very much for your advice. There was no missing data at baseline in our study. Since the study didn’t have so many follow-up factors, only FRS-R score, whether get to the endpoint. So actually, there was no missing data in our study. Thank you.

3. In the results section, the authors stated that survival time of the ALS-bvFTD group was significantly shorter than that of the pure ALS group. However, since the distribution was likely not normal, the Wilcoxon test or other approaches could be used.

Reply: Thank you very much for your kind advice. Yes, it was not normal distributed, and we have change into Mann-Whitney U test. The data has been changed as following: “The survival time of the ALS-bvFTD group was significantly shorter than that of the pure ALS group (z= 2.72, p=0.002) or the ALS with cognitive and behavioural changes group (z = 2.66, p =0.008) (Fig 3).” Thanks again!

4. The difference between pure ALS and cognitive impairment ALS was small. However, several baseline characteristics were needed to adjusted and log-rank test was need to be considered here.

Reply: Thank you very much for your suggestion! We tried Log Rank test to do survival analysis in pure ALS group and ALS with cognitive and behavioural change group, there was no significance (p=0.79). We have added in the result part, as following:” Log Rank test was used to compare the survival curve of pure ALS group and ALS with cognitive and behavioural change group, there was no significance (p=0.79). ”

5. The linear regression analysis between progression rate and follow-up time was performed. However，it might be nonlinear regression model and cubic or GEE model were worth be explored.

Reply: Thank you very much for your kind suggestion. According to your advice, we have tried other models as you mentioned, like cubic, while the results were similar: It still has significance of the correlation between ECAS score and survival time ( F=3.65, p=0.02). However, ECAS score have no correlation with FRS-R score at enrolment, the FRS-R progression rate from onset to enrolment, or the progression rate from enrolment to endpoint (p > 0.05). So we add this sentence in the manuscript :” The results were similar when using cubic and other regression models.” Thanks again!

---

## [Editor Report · Decision Letter 2]

2 Jun 2021

Prognosis of amyotrophic lateral sclerosis with cognitive and behavioural changes based on a sixty-month longitudinal follow-up

PONE-D-20-39850R2

Dear Dr. Fan,

We’re pleased to inform you that your manuscript has been judged scientifically suitable for publication and will be formally accepted for publication once it meets all outstanding technical requirements.

Kind regards,

Weidong Le

Academic Editor

PLOS ONE
---

## [Editor Report · Acceptance letter]

11 Jun 2021

PONE-D-20-39850R2 

Prognosis of amyotrophic lateral sclerosis with cognitive and behavioural changes based on a sixty-month longitudinal follow-up 

Dear Dr. Fan:

I'm pleased to inform you that your manuscript has been deemed suitable for publication in PLOS ONE. Congratulations! Your manuscript is now with our production department. 

Kind regards, 

on behalf of

Dr. Weidong Le 

Academic Editor

PLOS ONE